# The rDNA Diversity, Interseasonal Dynamic, and Functional Role of Cyanobacteria *Synechococcus* in the Sub-Arctic White Sea

**DOI:** 10.3390/plants13223153

**Published:** 2024-11-09

**Authors:** Tatiana A. Belevich, Irina A. Milyutina, Olga V. Vorob’eva, Aleksey V. Troitsky

**Affiliations:** 1Biological Faculty, Lomonosov Moscow State University, 119234 Moscow, Russia; olvorobieva@rambler.ru; 2Belozersky Institute of Physico-Chemical Biology, Lomonosov Moscow State University, 119992 Moscow, Russia; iramilyutina@yandex.ru; 3Russian Federation Research Institute of Fisheries and Oceanography, 105187 Moscow, Russia

**Keywords:** *Synechococcus*, cyanobacteria, picoplankton, 16S rRNA, ITS rDNA, phylogeny, White Sea, primary production

## Abstract

Planktonic unicellular cyanobacteria are the dominant biomass producers and carbon fixers in the global ocean ecosystem, but they are not abundant in polar seawater. The interseasonal dynamics of picocyanobacterial (PC) abundance, picophytoplankton primary production, and phylogenetic diversity of PC *Synechococcus* were studied in the sub-Arctic White Sea. The PC abundance varied from 0.2–0.3 × 10^6^ cells/L in February to 5.2–16.7 × 10^6^ cells/L in July. Picophytoplankton primary production ranged from 0.22 mg C/m^3^ per day in winter to 11.32 mg C/m^3^ per day in summer. *Synechococcus* abundance positively correlated with water temperature and river discharge that increased in recent years in the White Sea. Phylogenetic analysis of the 16S rRNA gene and ITS region clone libraries from the White Sea and Barents Sea eDNA revealed picocyanobacterial sequences related to marine *Synechococcus* subclusters 5.1-I, 5.I-IV, 5.2, and 5.3. All *Synechococcus* S5.1-I were common in the White and Barents seas and were consistently present in the picophytoplankton composition throughout the year. *Synechococcus* S5.2 and S5.3 appear in the PC community in summer, suggesting their river origin, and *Synechococcus* S5.1-IV inhabits only the Barents Sea and was not detected in the White Sea. A unique *Synechococcus* phylotype was revealed. It is expected that the increase in the abundance of PC and their increasing role in ecosystem functioning, as well as the enrichment of the species composition with new phylotypes in the semi-enclosed sub-Arctic White Sea, which is vulnerable to the effects of climate change, will be characteristic of all Arctic seas in general.

## 1. Introduction

Cyanobacteria are a globally widespread and diverse group of prokaryotes with a major impact on aquatic and terrestrial ecosystems of our planet. Photosynthetic cyanobacteria are responsible for a large proportion of global primary production and contribute to global carbon fixation by sequestering atmospheric carbon dioxide to mitigate climate change [1,2,3,4]. Moreover, nitrogen-fixing cyanobacteria significantly contribute to the biogeochemical cycling of nitrogen [5]. Small unicellular PC (cell size < 2 µm) together with photosynthetic picoeukaryotes belong to picophytoplankton. These tiny phototrophic prokaryotes are highly abundant in all oceans and are dominated by two genera, *Prochlorococcus* and *Synechococcus* [2,6]. According to the newest treatment of cyanobacteria, these microalgae belong to sister families Prochlorotrichales and Synechococcales, with 2 and 28 species, respectively [7]. However, the real diversity of these picoalgae greatly exceeds the number of formally described species. Many uncultivated cyanobacterial phylotypes have been detected in a number of eDNA studies, mostly of rDNA, in various marine and freshwater reservoirs in both oligotrophic and eutrophic habitats around the world [8,9,10,11,12]. However, at the same time, high-latitude regions remain underexplored [13,14,15].

Based on rDNA phylogeny, marine *Synechococcus* strains were classified into three major groups, termed subclusters 5.1, 5.2, and 5.3 [16]. Each of these subclusters is further detailed into several clades [17,18,19]. The community composition of PC is strongly influenced by environmental factors. *Synechococcus* is commonly found in high latitude cold waters in low abundance [15,20]. The distribution of *Prochlorococcus* is restricted to temperate, subtropical, and tropical waters. Recently, *Prochlorococcus marinus* was identified in the polar waters of the Spitsbergen shelf, Fram Strait [21]. The authors linked this fact to the process of Atlantification of the Arctic, which involves increased transport of Atlantic water masses through the Fram Strait towards the Arctic Ocean.

The White Sea adjoins the Barents Sea to the south of the Kola Peninsula. It is a marginal subpolar shelf region basin and has features similar to those of the Arctic shelf seas—presence of ice cover for 5–6 months, from December to May; constant negative temperature of the deep water mass and of the surface water in the winter; low illumination and polar night in the winter in most of the sea [22]. On the other hand, in terms of the temperature of the surface layer in summer, the White Sea refers to temperate seas. Thus, the White Sea is a unique marine environment, combining features of temperate and Arctic seas with the strong seasonality in temperature. Recent studies using metagenomic analyses revealed the presence of genetically distinct and rich pico-sized microbial populations fully adapted to the complex environment of the White Sea [23,24,25,26].

Previous studies of picophytoplankton communities in this basin have focused on the abundance and spatial distribution of picoalgae as a whole and on the species composition of phototrophic picoeukaryotes in ice and summer plankton, while the diversity of PC has been overlooked.

The White Sea is characterized by a high rate of water exchange with the Barents Sea. This has a significant impact on the entire marine ecosystem of the sea, including the phytoplankton. The species richness of large phytoplankton in the White Sea is only slightly lower than in the Barents Sea [27]. The discharge of the three major rivers, Severnaya Dvina, Onega, and Mezen, also significantly affects the phytoplankton communities [28]. Thus, it can be assumed that cyanobacterial diversity in the White Sea is determined by the influx of transformed Barents Sea water and river discharge. Rapid climate warming could lead to an increase in PC abundance in polar regions and an increase in *Synechococcus* diversity due to the appearance of temperate species.

The aims of the present study were to investigate the interseasonal diversity of natural communities of the picocyanobacterium *Synechococcus* by construction and sequencing of clone libraries of rDNA internal transcribed spacer (ITS) sequences and 16S rDNA, and to assess PC occurrence, biomass, and picophytoplankton primary production, as well as the contributions of PC to total picophytoplankton abundance in summer, autumn, and winter, and in the seasonal ice of the White Sea.

## 2. Materials and Methods

### 2.1. Water and Ice Sample Collection

The sampling was carried out in June 2014, July 2015, September 2016, and February 2017 in the Kandalaksha Bay (the White Sea) in the vicinity of the White Sea Biological Station, Lomonosov Moscow State University (66°33′ N, 33°06′ E) (Appendix A). In June 2015, seawater samples were collected during two cruises of the R.V. “Ecolog” in the Kandalaksha and Onega bays (Table 1).

Samples for genetic analysis were also collected at two stations in the central and northern parts of the Barents Sea during the 68th cruise of the R.V. ‘Akademik Mstislav Keldysh’ in July and August 2017 (Table 1).

Seawater was collected by 5 L Niskin-type bottles; a total of 21 samples were collected at 11 stations, 3 of which also collected ice cores. A titanium manual ice corer (14 cm of diameter) was used to collect ice cores and under-ice water. Sampling was carried out from two horizons 0 and 5 m.

The water and under-ice water salinity and temperature, and melted ice salinity measurements were obtained with a conductivity probe (Cond 3150i, WTW, London, UK).

### 2.2. Chlorophyll a (Chl-a) Concentration

Water samples (500 mL) for the total Chl-*a* concentrations (Chl_tot_) were filtered through Whatman GF/F filters (47 mm). A Turner Designs Model Trilogy field fluorometer was used to measure the Chl-*a* concentrations. The fluorometer was calibrated by a chemically pure chlorophyll solution (Sigma, Kawasaki, Japan) as the standard. The Chl-*a* concentrations were calculated using the standard equations [29]. To determine the Chl-*a* concentrations in the ice immediately after complete ice melt, 500–1000 mL subsamples were filtered through Whatman GF/F filters, then extractions and calculations were made according to the procedure for water samples.

Water samples (1000 mL) for the picophytoplankton Chl-*a* concentration (Chl_pico_) were successively filtered by a gentle reverse filtration system through Nucleopore filters with a pore size of 3 µm and then precipitate on Whatman GF/F filters.

### 2.3. Enumeration of Picocyanobacteria and Total Picophytoplankton Abundance and Biomass

For the enumeration of cyanobacteria and picoeukaryotes, an epifluorescence microscope (Leica DM2500, St. Gallen, Switzerland) was used. The subsamples (10 mL) were placed in a filtration funnel and incubated for 5–7 min after the saturated solution of primulin was added. Each sample was preserved with glutaraldehyde at a final concentration of 1%. Nuclear filters (0.12 m-pore diameter, Dubna, Russia) prestained with Sudan black were used for filtration. The cells on the filter were counted under a Leica DM1000 epifluorescence microscope at ×100, ×10, and ×1.3 magnifications. Depending on the cell concentration, 30 to 50 fields were examined, and the cell size was measured. For cyanobacteria, for which cell sizes varied from 0.8 to 1.2 µm (average 1 µm), a conversion factor of 470 ƒg C/cell was used [30]. The carbon biomass of picoeukaryotes was estimated according to a conversion factor of logC = 0.941 logV –0.60 [31], where V is the cell volume calculated as the volume of the relevant geometric bodies [32].

The average abundance and biomass were calculated for the 5 m depth of the stations for water samples and for the ice—per liter of melt water.

### 2.4. Primary Production (PP) and Growth Rate

PP was measured in the surface layer in July, September, and February (water). PP was estimated using radiocarbon modification of the light and dark bottle method [33]. Acid-cleaned 200 mL bottles with water samples after the addition of sodium bicarbonate (NaH^14^CO_3_, 0.05 µCi per 1 mL of sample) were exposed for 4–6 h of a light day in situ conditions. After exposure, the samples were filtered through a 3 µm pore size filter onto a 0.45 µm nitrocellulose filter. The total PP values were obtained by summation of large phytoplankton fraction (>3 µm) and picophytoplankton (<3 µm) PP values. After filtration, the samples were treated with 0.1 N HCl and filtered seawater, dried, and frozen until laboratory processing. Filters were placed in a scintillation vial with 10 mL of scintillation cocktail (Optiphase HiSafe III, PerkinElmer, Waltham, MA, USA). The samples’ radioactivity was determined using a liquid scintillation counter, RackBeta 1215 (LKB Wallac, Turku, Finland).

Picophytoplankton specific growth rates (μ, d^–1^) in the surface layer were calculated from surface primary production and biomass values assuming exponential growth:μ = ln [(B + PP_pico_)/B_pico_],
where B_pico_ is the carbon picophytoplankton biomass (mg C/m^3^) and PP_pico_ is the picophytoplankton primary production (mg C/m^3^ per day).

### 2.5. DNA Extraction, PCR Amplification, Cloning, and Sequencing

Twenty samples were analyzed for the investigation of the PC diversity (Table 1). For DNA isolation, 3–5 L of water samples were filtered through a 3 µm pore size polycarbonate filter using a <50 mm Hg vacuum. The filtrate (<3 µm) was then filtered through 0.2 µm Sterivex units (Millipore Canada Ltd., Mississauga, ON, Canada). The buffer was added to the Sterivex units (1.8 mL of 50 mM Tris–HCl, 0.75 M sucrose, and 40 mM EDTA; pH 8.3), and the samples were stored at –80 °C until nucleic acid extraction. Genomic DNA was extracted using the NucleoSpin Plant kit (Macherey-Nagel, Düren, Germany) following the manufacturer’s instructions.

The ITS region of rDNA was amplified with the cyanobacterium-specific primers Picocya16S-F—Picocya23S-R [34]. Amplification conditions were described in Belevich et al., 2023 [15]. PCR products were cloned into pAL2-T vectors using a Quick-TA kit (Evrogen, Moscow, Russia) in accordance with the manufacturer’s instructions. Vector-specific primers M13F and M13R were used to screen clones. A total of 218 ITS clones were sequenced. Summer samples 2/15 and 6/15 demonstrated the highest diversity of ITSs, representing all three known cyanobacteria subclusters 5.1, 5.2, and 5.3. To obtain partial sequences of the 16S rRNA gene (1163 nucleobases), DNA from these samples was amplified with primers Cya359F [35] and Picocya23S-R. The obtained PCR products contained a partial sequence of the 16S rRNA gene, and ITSs were cloned according to the same protocol used for the ITS. A total of 7 clones with ITSs from different subclusters were selected to construct the phylogenetic tree. DNA sequencing was performed using the ABI PRISM^®^ BigDye™ Terminator v. 3.1 kit, followed by analysis of the reaction products on an Applied Biosystems 3730 DNA Analyzer (Waltham, MA, USA) in the Genome Center (Engelhardt Institute of Molecular Biology, Russian Academy of Sciences, Moscow, Russia).

### 2.6. Phylogenetic Analysis

The obtained sequences were compared to GenBank entries using BLAST (http://www.ncbi.nlm.nih.gov/BLAST/ (30 July 2024)) in order to identify closely related sequences in the database. Two datasets were arranged: ITS and 16S rDNA + ITS. Sequences were aligned using the Muscle5 [36] with the default setting, followed by manual adjustment in BioEdit ver. 7.2.5 [37]. The tRNA-Ile and tRNA-Ala gene sequences were removed from the alignment before the phylogenetic trees were constructed. From 218 sequenced ITS clones, 31 sequences differing by at least 0.15% were selected and used for phylogenetic analysis together with 40 homologous sequences from GenBank. The 16S rDNA + ITS dataset consists of 24 entries, 7 of which are new.

Sequences obtained from this study were deposited in GenBank (http://www.ncbi.nlm.nih.gov) under accession numbers PQ360487-PQ360493 for 16S rRNA and PQ360496-PQ360713 for the ITS.

Phylogenetic trees were computed by the maximum likelihood (ML) approach realized in IQ-TREE ver.2.2.0 http://www.iqtree.org/ [38] using 1000 ultrafast bootstrap replicas [39] and the minimum evolution (ME) method [40] implemented in MEGA ver.11.0.13 [41] using 1000 bootstrap replicas. The best substitution model for the ML tree search was chosen by ModelFinder [42] in IQ-TREE. The evolutionary distances as the number of base substitutions per site in the ME trees were computed using the maximum composite likelihood method. The *p*-distances for the ITS sequences were calculated in MEGA.

## 3. Results

### 3.1. Environmental Parameters

The highest surface temperature was recorded in July, averaging 16.3 °C; the lowest temperature was recorded in February and varied from −1.4 to 1.1 °C (Figure 1a). Surface salinity was practically the same in July and September, averaging 24.8 psu and 24.3 psu, respectively (Figure 1b). In June, the average salinity was 20.4 psu and varied widely. Two samples (2/15 and 6/15) were collected near the areas of the freshwater discharge influence from the channel of Knyazhaya hydroelectric power station and Onega River (Appendix A). The ice temperature ranged from −2.1 to −2.9 °C; the ice salinity averaged 6.7 psu.

The concentration of Chl_tot_ (Figure 1c) varied from 0.4 µg/L to 4.2 µg/L in June, from 0.71 to 1.2 µg/L in September, and from 0.02 to 0.04 µg/L in February in under-ice water. In ice, this parameter was higher than in water and averaged 0.5 µg/L.

### 3.2. Picocyanobacteria Abundance and Role in Total Picophytoplankton

*Synechococcus*-like PC were found in every season. The lowest PC abundance was in the under-ice water and ranged from 0.2 to 0.3 ×10^6^ cells/L; the highest abundance was in July and varied from 5.2 to 16.7 ×10^6^ cells/L (Figure 1d and Figure 2). The average PC biomass in under-ice water was 0.12 µg C/L in July—4.78 µg C/L.

PC always represented a high percentage of the picophytoplankton cells (Figure 2). The contribution of PC to the total picophytoplankton biomass was 75% and 85% in under-ice water and ice, respectively. The proportion of PC reached 97% in July and decreased to 77% in September.

In general, picophytoplankton represented on average 43% of the total phytoplankton biomass in September and ~50% in February for under-ice water, but only 27% in July. The phytoplankton community contained less than 5% of picophytoplankton in June and winter ice (Appendix A).

### 3.3. Primary Production of Total Phytoplankton and Picophytoplankton

The primary production of total phytoplankton (PP_tot_) varied significantly in different seasons (Table 2). The highest PP_tot_ was found in July with polar day and high-water temperature, and the lowest—in winter, in under-ice water, when daylight duration did not exceed 6 h. The same trend was found for picophytoplankton primary production (PP_pico_)—it decreased from July to February. However, the relative contribution of PP_pico_ to PP_tot_ increased from 15% in summer to 63% in the winter season. Mean picophytoplankton specific growth rate was the highest in autumn; in winter it was below 1 day^−1^ (Table 2).

### 3.4. Synechococcus Molecular Diversity Using ITS and the 16S rRNA Gene

A total of 179 ITS clones were sequenced from the White Sea samples, ranging from 39 to 80 for each season. In addition, 40 clones were sequenced from the Barents Sea samples. The ITS length among environmental clones varied widely, from 754 to 988 bp. The ITS alignment of 1328 positions consists of 625 parsimony-informative sites, 112 singletons, and 591 constant sites. The best-fit evolutionary model K3Pu + F + I + G4 was chosen for the tree search according to BIC. Phylogenetic analysis revealed three major highly supported clades with ultrafast bootstrap values (BVs) = 84–100% corresponding to marine *Synechococcus* subclusters 5.1, 5.2, and 5.3 (Figure 3). ML and ME trees are mostly topologically congruent, with the exception of the displacement of two branches marked by red arrows in Figure 3 and the branching order in clade 5.1-I consisting of very similar sequences with *p*-distances < 0.04 (Appendix A).

The 16S rDNA + ITS alignment of 24 sequences consists of 2387 positions, of which 578 are parsimony-informative, 143 are singletons, and 1666 are constant. The best-fit evolutionary model TVM + F + I + G4 was chosen for the tree search according to BIC. The bootstrap consensus ML tree is shown in Figure 4. On this tree, subclusters 5.1 and 5.3 are monophyletic with maximum support, and subcluster 5.2 is paraphyletic and organized in two clades, one of which, supported by 100%, is sister to subcluster 5.1 with 82% support.

A total of fourteen phylotypes were identified, nine of which are represented by different *Synechococcus* (Figure 5). The similarity of one of them to known sequences from the GenBank was less than 97% (~84%, HQ724261). Five phylotypes are of uncertain genus and belong to uncultured cyanobacteria. The similarity between sequences of the same phylotype was more than 99%. Sequences from different phylotypes of subcluster 5.1 had similarity less than 98%, in most cases 95–97%. In the other two subclusters, the similarity of phylotypes was significantly lower and amounted to 70–80%.

Two phylotypes of uncultured *Synechococcus* sp. (HQ723628 and MW604308) were predominant across all stations and were found in all seasons in the White Sea and in the summer plankton of the Barents Sea (Figure 5). *Synechococcus* sp. (OR448690), previously revealed in the Kolyma estuary (the East Siberian Sea), was also widespread and found in the Barents Sea and in all seasons except autumn in the White Sea. These three commons for both sea phylotypes belong to marine subcluster S5.1-I (Figure 3). Moreover, one more *Synechococcus* S5.1-IV (MG668503) was found only in the Barents Sea and was completely absent in the White Sea phytoplankton, as was *Synechococcus* S5.1-I (HQ723592).

One uncultured cyanobacterium clone (OR448736) previously discovered in the Laptev Sea was found in the summer plankton of the Barents and White Seas. Phylogenetic analysis showed that this phylotype belongs to subcluster 5.2. Seven phylotypes were detected only in the summer plankton of the White Sea and were absent in the Barents Sea; five of them belong to subcluster 5.2, and the least two—FJ596205, HQ724261—referred to subcluster 5.3. All these “summer” phylotypes were found in samples carried out at freshened areas (stations 2/15 and 6/15, Table 1). One more *Synechococcus* (JF306763) was detected only in the autumn plankton of the White Sea and in the ice.

## 4. Discussion

Our studies show that PC play an important role in the phytoplankton of the sub-Arctic White Sea. They dominate the picophytoplankton composition in all seasons. PC abundance was 12% higher in July 2014 than in July–August 2009 (8.9 × 10^6^ cells/L) [43]. PC abundance in February 2017 was 29% higher than that in April 2010 in ice and five times higher in under-ice water (0.4 × 10^6^ cells/L and 0.05 × 10^6^ cells/L) [43]. The increase in abundance compared to previous studies is obviously related to the fact that one of the main sources of PC input to the Arctic marine environment is river runoff [13,20,44]. In the White Sea, the runoff of the Northern Dvina, Kem, Nizhny Vyg, Onega, Mezen, Ponoi, and Kovda rivers increased by an average of 11% for the period 1955–2019 [45], and the water temperature at the mouths of the rivers flowing into the White Sea increased by more than 1 °C for the period 1956–2015 [46]. In addition, during 2004–2020 the average duration of the ice season decreased by 10 days and the average values of ice cover decreased by 3–7% [47], the average long-term air temperature at the coastal stations of the White Sea increased by 1.4 °C (http://meteo.ru). Water temperature is one of the main factors influencing the distribution and growth rates of *Synechococcus* in the marine environment [48,49,50]. *Synechococcus* abundance in surface water showed a positive correlation with water temperature in the White Sea (R = 0.85; *p* < 0.0001). Under current global warming trends, which are amplified in the Arctic, a further increase in the proportion of PC, and especially *Synechococcus*, in the phytoplankton of sub-Arctic seas is expected [2].

In July, picophytoplankton biomass and PP_pico_, as well as biomass and production of micro- and nanophytoplankton, reach the summer peak. This is consistent with data on the seasonal dynamics of phytoplankton in the White Sea [28]. The values of PP_pico_ in summer were comparable with the data obtained in the summer of 1998 with the dominance of cyanobacteria (16.8 mg C/m^3^ per day) [51]. Estimates of PP_pico_ in autumn and winter were obtained for the first time. At the same time, the contribution of PP_pico_ to PP_tot_ in summer and autumn was low and practically did not differ. The higher maximum growth rate of picophytoplankton in autumn was related to the proportion of picoeukaryotes in the total abundance: eukaryotic cells outgrew cyanobacteria and contributed a larger fraction of total carbon fixation by picophytoplankton [52]. In winter, despite the relatively high proportion of picoeukaryotes in the picophytoplankton, the growth rate was minimal. This appears to be due to the low water temperature, which reduces growth rate [53,54].

All *Synechococccus* phylotypes common for the White and Barents seas related to marine subcluster 5.1-I are consistently present in the picophytoplankton composition throughout the year. This subcluster, largely confined to coastal and higher latitude regions (above 30°N or below 30°S approximately) [55], exhibits great tolerance to cold temperatures and prefers high nutrient levels [56]. It is the waters of the Barents Sea that are a constant source of *Synechococcus* from this subcluster entering the White Sea. The adaptation of the *Synechococcus* S5.1-I to low temperatures makes them the only representatives of autumn picophytoplankton and inhabitants of ice and under-ice waters with temperatures below 0 °C. *Synechococcus* S5.1-I, together with *Synechococcus* S5.1-IV earlier, were retrieved as the only ones inhabiting the eastern part of the Fram Strait, where Atlantic water is transported northward by the West Spitsbergen Current in all seasons [57]. *Synechococcus* S5.1-IV was clearly dominating in Atlantic waters, while S5.1-I appeared to dominate in colder Arctic waters. Over the course of our study, phylotypes of both clades were found in the Barents Sea, whose waters are formed by transformed Atlantic waters and Arctic waters. *Synechococcus* S5.1-IV phylotypes were not revealed in the White Sea even in summer, which was possibly due to the relatively low salinity of the waters of the White Sea compared to the Barents Sea. Possibly, phylotypes of *Synechococcus* S5.1-IV have a low tolerance to wide salinity ranges. The tendency of decreasing *Synechococcus* concentrations with decreasing salinity was noted [57].

In summer, cyanobacteria diversity is greatly enhanced by phylotypes of subclusters 5.2 and 5.3. *Synechococcus* S5.2 usually inhabits estuarine and freshwater environments and contains mainly euryhaline strains that are better adapted to a wider range of salinities than other *Synechococcus* lineages [10,58]. Whereas members of S5.3 were mostly detected in the surface water of open-ocean habitats in the northwestern Atlantic and Pacific Ocean [17,18]. *Synechococcus* subcluster 5.3 was only re-established recently, and to date it was suggested that it includes at least six clades (5.3-I to 5.3-VI) [59]. Our data show that the diversity of *Synechococcus* S5.3 has not been fully recognized due to the lack of data from Arctic and sub-Arctic seas and that more diversity of S5.3 should be expected. The adaptation of newly discovered phylotypes of *Synechococcus* S5.3 to temperature, salinity, or light conditions requires future studies.

We assume that river discharge is the main source of *Synechococcus* S5.2 and S5.3 in summer. Relatively high surface water temperatures, low salinity, and favorable light conditions during the polar day allow the phylotypes of these two subclusters to grow in the White Sea. With the onset of autumn, when the water temperature decreases and the length of daylight is reduced to almost zero, only *Synechococcus* S5.1-I can survive. Our results confirm advective transport from the Barents Sea as one of the sources of PC in the White Sea. Phylotypes of *Synechococcus* S5.1-I are well adapted to cold temperatures and low light, but also to sub-Arctic summer conditions with water surface temperatures above 10 °C.

We found only one unique *Synechococcus* phylotype with 84% similarity to previously reported sequences. This may be due to the rare occurrence of unique phylotypes, and further studies of PC diversity using a metagenomic approach and other marker genes, for example rpoC1, are needed.

## 5. Conclusions

The semi-enclosed sub-Arctic White Sea is uniquely vulnerable to the consequences of climate change. Warming sea surface temperature and increasing river discharge lead to cyanobacterial growth, especially in the autumn–winter period when the abundance of micro- and nanophytoplankton decreases. It is precisely in winter that picophytoplankton, 85% of which are cyanobacteria, produce more than 60% of the total primary production. Due to the primary production of picoforms in winter, microzooplankton can lead a quite “well-fed” existence during the low productivity season [60]. This is confirmed by the high taxonomic richness and abundance of small heterotrophs identified during this period [61]. Seasonal variations in PC composition were determined by two sources: the Barents Sea waters and river discharge. The cold waters of the Barents Sea determined the constant PC composition of the White Sea, consisting of *Synechococcus* S5.1-I with a high tolerance to cold temperatures. In summer, riverine inputs enriched the PC community with thermophilic and low salinity phylotypes, predominately *Synechococcus* S5.2 and 5.3.

## Figures and Tables

**Figure 1 plants-13-03153-f001:**
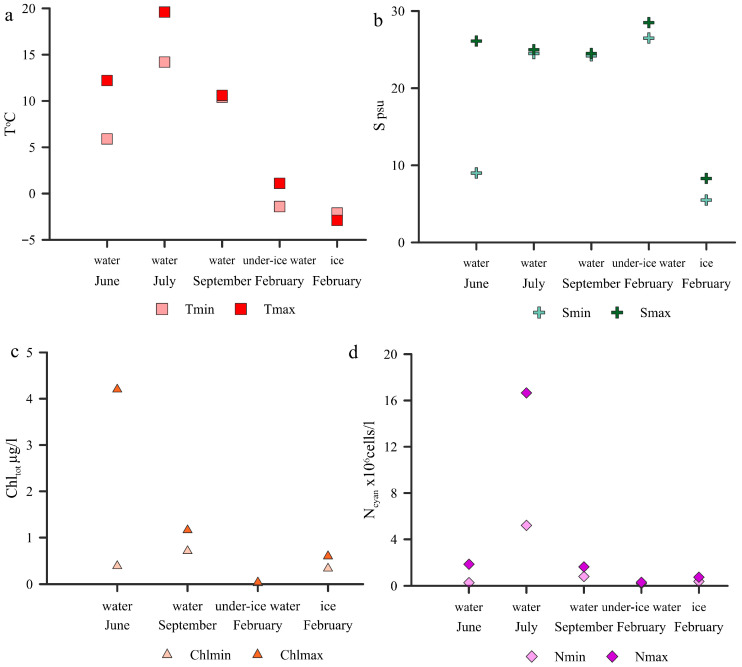
Variation in minimum (light symbols) and maximum (dark symbols) surface temperature (**a**), salinity (**b**), total chlorophyll “a” concentration (**c**), and picocyanobacteria abundance (**d**) in different seasons in the White Sea.

**Figure 2 plants-13-03153-f002:**
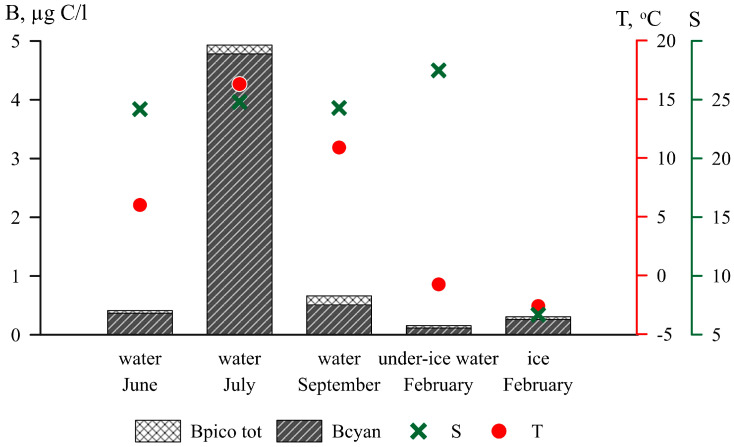
Total picophytoplankton biomass (B_pico tot_, µg C/L), picocyanobacteria biomass (B_cyan_, µg C/L), average water temperature (T °C), and salinity (S, psu) in June, July, September, and February in the White Sea.

**Figure 3 plants-13-03153-f003:**
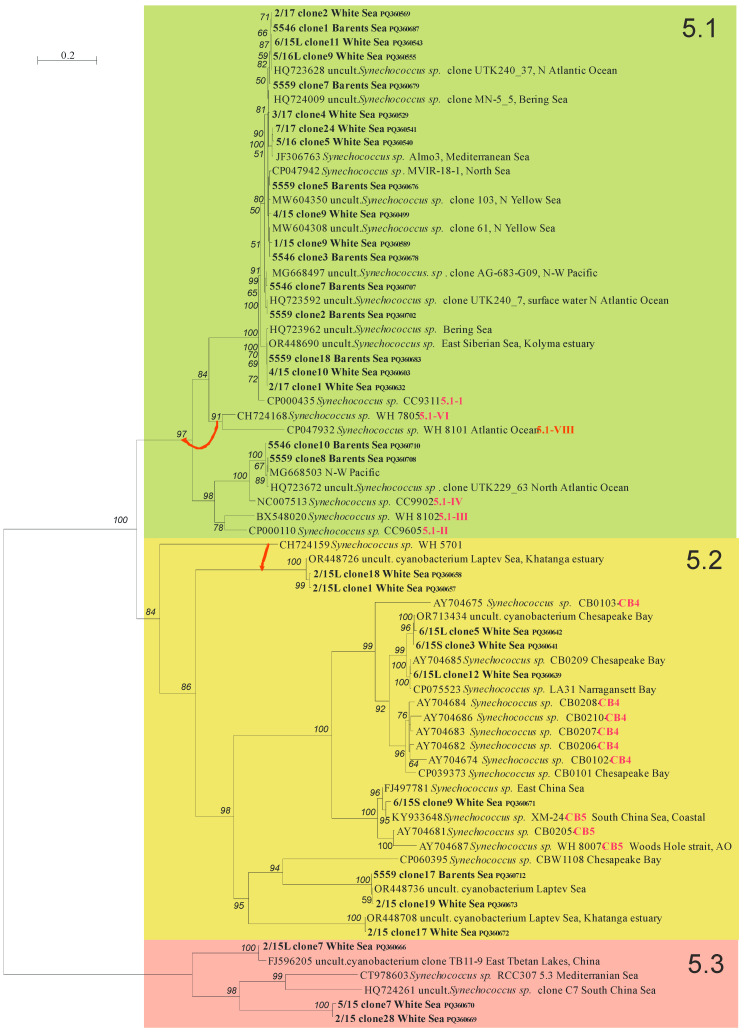
The ML phylogenetic tree of *Synechococcus* strains based on ITS rDNA sequences. BV in percentages higher than 50% are shown. Two red arrows show the position of branches on the ME tree. Subclusters 5.1, 5.2, and 5.3 are marked by different colors. Clones obtained in this study are marked bold. The scale bar indicates the number of nucleobase substitutions per site.

**Figure 4 plants-13-03153-f004:**
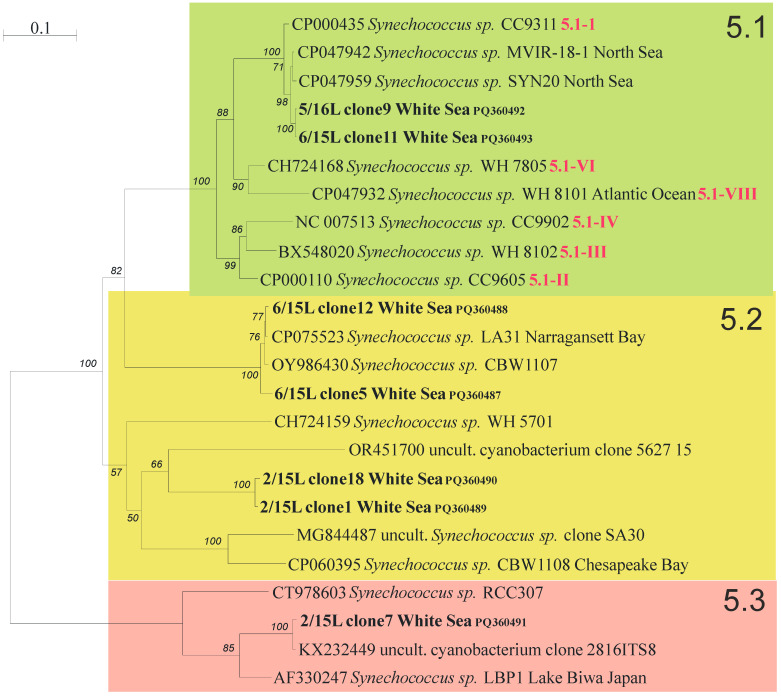
The ML phylogenetic tree of *Synechococcus* strains based on 16S rDNA + ITS sequences. BV in percentages higher than 50% are shown. Subclusters 5.1, 5.2, and 5.3 are marked by different colors. Clones obtained in this study are marked bold. The scale bar indicates the number of nucleobase substitutions per site.

**Figure 5 plants-13-03153-f005:**
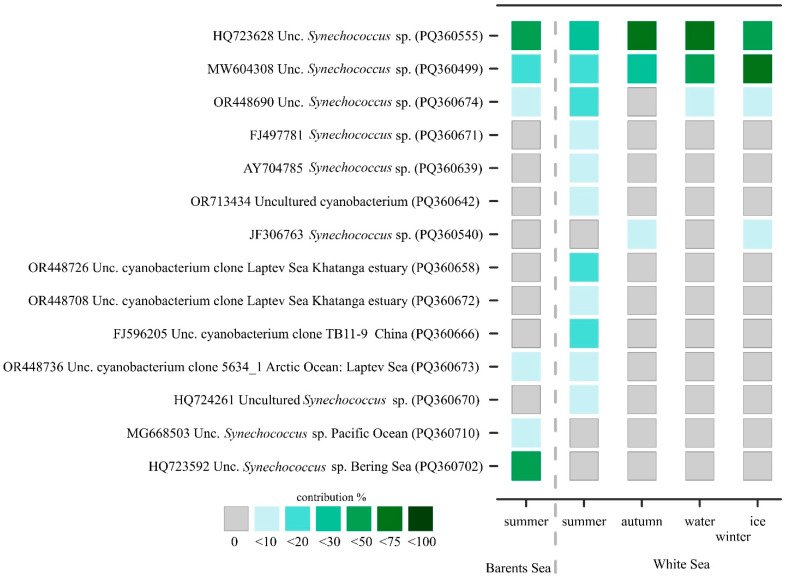
*Synechococcus* community composition across the studied seas. The heatmap reflects the contribution of phylotypes as a percentage of the total clones. Representative clones from different phylotypes are presented.

**Table 1 plants-13-03153-t001:** Location of sampling stations, date and horizon of sampling, sample type, surface temperature (T_0_ °C) and salinity (S_0_ psu), total chlorophyll a concentration (Chl_tot_, µg/L), and picophytoplankton chlorophyll a concentration (Chl_pico_, µg/L) at the sampled horizon. * No DNA, - no data.

Sample	Latitude (N)	Longitude (E)	Bay	Date, Horizon	Water/Ice	T_0_	S_0_	Chl_tot_	Chl_pico_
The White Sea									
1/15	66.870	32.804	Kandalaksha	12 June 2015 0 m	water	5.4	19.9	2.63	-
2/15	66.893	32.468	Kandalaksha	12 June 2015 0 m	water	6.3	9.0	1.50	-
3/15	66.531	33.726	Kandalaksha	17 June 2015 0 m	water	5.9	25.5	4.20	-
4/15	66.535	33.721	Kandalaksha	17 June 2015 250 m	water	5.9	25.5	0.04	-
5/15	65.048	35.257	Onega	18 June 2015 0 m	water	6.6	26.1	0.72	-
6/15	64.349	37.062	Onega	24 June 2015 0 m	water	12.2	16.2	0.39	-
21/14	66.553	33.105	Kandalaksha	18 July 2014 0 m	water	13.4	24.5	-	-
22/14	66.536	33.220	Kandalaksha	21 July 2014 0 m	water	14.0	24.5	-	-
* 23/14	66.581	32.982	Kandalaksha	25 July 2014 0 m	water	19.6	24.5	-	-
1/16	66.525	33.093	Kandalaksha	13 September 2016 0 m	water	10.6	24.2	0.77	-
2/16	66.536	33.220	Kandalaksha	13 September 2016 0 m	water	10.4	24.3	1.16	-
5/16	66.552	33.035	Kandalaksha	15 September 2016 0 m	water	10.5	24.5	0.71	0.16
2/17	66.534	33.109	Kandalaksha	4 February 2017 0 m	under-ice water	−1.3	26.5	0.03	0.005
3/17	66.581	32.982	Kandalaksha	5 February 2017 0 m	under-ice water	−1.4	28.5	0.04	0.003
4/17	66.534	33.109	Kandalaksha	4 February 2017	ice	−2.1	8.3	0.56	0.015
5/17	66.581	32.982	Kandalaksha	5 February 2017	ice	−2.9	6.2	0.34	0.01
6/17	66.552	33.035	Kandalaksha	6 February 2017 0 m	under-ice water	−1.4	26.6	0.04	0.01
7/17	66.552	33.035	Kandalaksha	6 February 2017	ice	−2.8	5.5	0.6	0.02
8/17	66.553	33.105	Kandalaksha	3 February 2017 0 m	water	1.1	28.2	0.02	0.003
The Barents Sea									
5546	72.549	27.346		31 July 2017, 5 m	water	8.8	34.8	0.68	-
5559	80.066	44.857		5 August 2017, 15 m	water	2.5	33.4	0.15	-

**Table 2 plants-13-03153-t002:** Phytoplankton primary production (PP_tot_, mg C/m^3^ per day), picophytoplankton primary production (PP_pico_, mg C/m^3^ per day), the contribution of PP_pico_ to PP_tot_, and picophytoplankton specific growth rates (μ, d^–1^) in the surface layer of the White Sea in different seasons.

Season	Month	PP_tot_	PP_pico_	% PP_pico_	µ
Summer	July	75.13	11.32	15	1.13
Autumn	September	17	2.15	13	1.22
Winter (water)	February	0.35	0.22	63	0.9

## Data Availability

The data generated or analyzed during this study are included in this published article and its Appendix A. Sequences obtained from this study were deposited in GenBank (http://www.ncbi.nlm.nih.gov) under accession numbers PQ360487-PQ360493 and PQ360496-PQ360713.

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
