# Peer review of "The rDNA Diversity, Interseasonal Dynamic, and Functional Role of Cyanobacteria *Synechococcus* in the Sub-Arctic White Sea"

_plants, 2024, doi:10.3390/plants13223153_

Round 1

Reviewer 1 Report

Comments and Suggestions for Authors

The article is devoted to the study of the cyanobacterial component of the plankton of the northern White Sea of the Russian Federation. The team of authors is qualified and has publications on this topic, their competence and ability to work in the conditions of the north does not cause doubts.

The introduction is complete and sufficient for understanding the essence of the text. The methods section is written clearly and exhaustively, allows to reproduce the methods and techniques. The techniques are up-to-date, tested and reliable and fully meet the authors' objectives and goals.

The text is written clearly and understandably, with the necessary detail to trace the logic and path of thought of the authors. I have a few comments, which I present here in line by line tabular:

line(s) comments

-----------------------------------------

39, 40 It seems appropriate to mention here carbon fixation and potential climate impacts of cyanobacteria and photosynthesizing plankton along with those roles listed above!

232, 238, 258  The units of seawater salinity, g/L, are missing both in the text and on the ordinate axis of the corresponding diagrams

238, 258, 329 Looking at the diagram one can't help but notice the poor quality of the picture, as well as poor readability of some captions. I hope the authors will take care of improving the quality of these pictures!

423 (General wishes) it seems to me that a logical extension of the work would be to attempt shotgun metagenomic sequencing, since obviously it might be of interest to compare genomics of winter survivors strains and of those unable to do so!

The work provides important ground for thinking about phylogeny and the ability of cyanobacteria to survive the harsh conditions of the far north. Such exploratory studies are very useful for both basic science and applied aspects of cyanobacteria utilization for human needs! Good luck to the authors in their work!

Author Response

We thank the Editor and the reviewers for their time and valuable comments which allowed us to improve the text of the article. We have prepared a revised version of the paper taking into account the comments. We also tried our best to rewrite the Materials and Methods to increase the originality of the text.

 Response to editor's comment

Dear Zoey Deng,

We have tried to overcome self-citation in Materials and Methods as best we could. In fact, it is not easy, since the information provided in this section is quite standard.

Point-by-point response to reviewer 1

The article is devoted to the study of the cyanobacterial component of the plankton of the northern White Sea of the Russian Federation. The team of authors is qualified and has publications on this topic, their competence and ability to work in the conditions of the north does not cause doubts.

The introduction is complete and sufficient for understanding the essence of the text. The methods section is written clearly and exhaustively, allows to reproduce the methods and techniques. The techniques are up-to-date, tested and reliable and fully meet the authors' objectives and goals.

The text is written clearly and understandably, with the necessary detail to trace the logic and path of thought of the authors.

The work provides important ground for thinking about phylogeny and the ability of cyanobacteria to survive the harsh conditions of the far north. Such exploratory studies are very useful for both basic science and applied aspects of cyanobacteria utilization for human needs!

Response: Thank you for your careful consideration of our manuscript and your favorable assessment of it.

39, 40 It seems appropriate to mention here carbon fixation and potential climate impacts of cyanobacteria and photosynthesizing plankton along with those roles listed above!

Response: We add a reference [4] on carbon recycling by cyanobacteria. The corrected text: “Photosynthetic cyanobacteria are responsible for a large proportion of global primary production, and contribute to global carbon fixation sequestered atmospheric carbon dioxide to mitigate climate change [1-4]”

 232, 238, 258 The units of seawater salinity, g/L, are missing both in the text and on the ordinate axis of the corresponding diagrams

Response: corrected. Seawater salinity is defined in practical salinity units (psu), where one gram of salt per 1000 grams of water is one psu. We have added the salinity dimension to the text and figures.

238, 258, 329 Looking at the diagram one can't help but notice the poor quality of the picture, as well as poor readability of some captions. I hope the authors will take care of improving the quality of these pictures!

Response: We changed Figures 1 and 2 in the text. The quality of the figures has deteriorated during processing of the submitted manuscript. All original drawings (>300 dpi) are provided as separate files.

423 (General wishes) it seems to me that a logical extension of the work would be to attempt shotgun metagenomic sequencing, since obviously it might be of interest to compare genomics of winter survivors strains and of those unable to do so!

Response: Thank you for your valuable advice. We plan to carry out such works in the White Sea next year.

Reviewer 2 Report

Comments and Suggestions for Authors

The manuscripton describes Synechococcus in the marginal ocean of the White Sea with regard to the interseasonal dynamics of picocyanobacterial abundance and productivity including the diversity of the various picophytoplankton communities.  In general the manuscript is well written and the English is very good.  The figures are easy to read and appropriate for the paper.  Unfortunately Table 2 is split between pages which probably occurred because of the length Figure 3.  The conclusions are a good summary of the main results of the research.  

Author Response

Point-by-point response to reviewer 2

In general, the manuscript is well written and the English is very good.  The figures are easy to read and appropriate for the paper.  Unfortunately, Table 2 is split between pages, which probably occurred because of the length Figure 3. The conclusions are a good summary of the main results of the research.

Response: Thank you for your thorough review and valuable feedback on our manuscript. We appreciate your positive comments regarding the writing and the description of the results.

The position of Table 2 is corrected.